

# Mobile acoustic transects miss rare bat species: implications of survey method and spatio-temporal sampling for monitoring bats

Elizabeth C. Braun de Torrez, Megan A. Wallrichs, Holly K. Ober and Robert A. McCleery

Department of Wildlife Ecology and Conservation, University of Florida, Gainesville, FL, United States of America

## ABSTRACT

Due to increasing threats facing bats, long-term monitoring protocols are needed to inform conservation strategies. Effective monitoring should be easily repeatable while capturing spatio-temporal variation. Mobile acoustic driving transect surveys ('mobile transects') have been touted as a robust, cost-effective method to monitor bats; however, it is not clear how well mobile transects represent dynamic bat communities, especially when used as the sole survey approach. To assist biologists who must select a single survey method due to resource limitations, we assessed the effectiveness of three acoustic survey methods at detecting species richness in a vast protected area (Everglades National Park): (1) mobile transects, (2) stationary surveys that were strategically located by sources of open water and (3) stationary surveys that were replicated spatially across the landscape. We found that mobile transects underrepresented bat species richness compared to stationary surveys across all major vegetation communities and in two distinct seasons (dry/cool and wet/warm). Most critically, mobile transects failed to detect three rare bat species, one of which is federally endangered. Spatially replicated stationary surveys did not estimate higher species richness than strategically located stationary surveys, but increased the rate at which species were detected in one vegetation community. The survey strategy that detected maximum species richness and the highest mean nightly species richness with minimal effort was a strategically located stationary detector in each of two major vegetation communities during the wet/warm season.

# INTRODUCTION

Effective methods to survey wildlife populations are critical to document changes in biodiversity and develop appropriate management actions. These surveys must be easily repeatable over time while still capturing the inherent variation in a population or community. However, surveys can be logistically complicated and costly, particularly in large or remote areas that are difficult to access (*Conn et al., 2016*). Bats (Order Chiroptera) are a taxonomic group that are challenging to survey due to their elusive nocturnal behavior,

Corresponding author
Elizabeth C. Braun de Torrez, ecbraun@ufl.edu

and ability to fly over vast landscapes. Bats provide important pollination, seed dispersal and pest suppression services (*Kunz et al., 2011*), yet the long-term provisioning of these services is uncertain due to sharp declines of bats worldwide from habitat loss, emerging pathogens and wind energy development (*Fisher et al., 2012*; *Hayes, 2013*; *Mickleburgh, Hutson & Racey, 2002*). Long-term monitoring protocols need to be developed to document bat population trends and inform conservation strategies (*O'Shea, Bogan & Ellison, 2003*).

Techniques to acoustically detect and identify bats have improved over the past several decades (*Britzke, Gillam & Murray, 2013*; *Parsons & Szewczak, 2009*), allowing documentation of species distributions, habitat preferences, activity patterns, and presence of endangered species (e.g., *Britzke et al., 2002*; *Jaberg & Guisan, 2001*; *Wickramasinghe et al., 2003*; *Williams, O'Farrell & Riddle, 2006*). However, it is still not clear how best to use acoustic surveys to monitor bat communities (*Hayes, Ober & Sherwin, 2009*). Specifically, we do not know the temporal and spatial sampling effort necessary to represent the heterogeneity in bat activity and species composition across landscapes (*Britzke, Gillam & Murray, 2013*; *Loeb et al., 2015*). Nightly variation in bat activity can result in misrepresentation of the species assemblage in a site if not sufficiently sampled temporally (*Hayes, 1997*; *Law et al., 2015*; *Milne et al., 2005*; *Skalak, Sherwin & Brigham, 2012*). Similarly, spatial heterogeneity can lead to different estimates of bat activity and species richness, even among acoustic detectors located within the same vegetation stand (*Britzke, 2003*; *Duchamp et al., 2006*; *Fischer et al., 2009*; *Froidevaux et al., 2014*). Recommendations for duration and spatial stratification of surveys have varied widely among studies (*Bean & Rowland, 1997*; *Britzke, Gillam & Murray, 2013*; *Coleman et al., 2014*; *Froidevaux et al., 2014*; *Rodhouse, Vierling & Irvine, 2011*; *Stahlschmidt & Brühl, 2012*).

Two primary acoustic survey methods are frequently recommended for use in bat monitoring protocols: stationary acoustic point surveys (hereafter 'stationary surveys'), and mobile acoustic driving transect surveys (hereafter 'mobile transects') (e.g., North American Bat Monitoring Program (NABat); *Loeb et al., 2015*). Stationary surveys and mobile transects each have their respective advantages (Table 1), and are often encouraged to be used together (*Loeb et al., 2015*). However, there are many cases in which limited funding, equipment and time may require biologists to select only one method. Despite the prevalent use of both stationary surveys and mobile transects, the relative efficacy of each method has rarely been compared for bats (but see *Whitby et al., 2014*). Mobile transects are considered by some to be a robust and cost-effective method to monitor bat populations (*Battersby, 2010*; *Catto, Russ & Langton, 2003*; *Roche et al., 2011*; *Whitby et al., 2014*). Because most bat species fly slower than the recommended speed of a survey vehicle (32 km/h) (*Hayward & Davis, 1964*; *Patterson & Hardin, 1969*), each recorded bat pass should represent a unique individual, allowing estimates of relative abundance (*Roche et al., 2011*) that cannot be derived from stationary detectors (*Hayes, 2000*). However, mobile transects may misrepresent patterns of bat community composition if use of roads or roadside habitat differs among species (*Anderson, 2001*; *Linton, 2009*; *Roche et al., 2011*; *Zurcher, Sparks & Bennett, 2010*). Of particular concern is that, globally, many large protected areas are remotely located with limited access to roads (*Joppa & Pfaff, 2009*), thereby restricting the potential areas in which mobile transects can sample the bat

**Table 1** **Comparison of the potential advantages and disadvantages associated with two primary acoustic survey methods, mobile driving transect surveys and stationary point surveys, recommended to survey bats (*Loeb et al., 2015*).** Stationary surveys are further separated into survey methods in which a single detector is strategically located ('single strategic') and multiple detectors are spatially replicated across the landscape ('spatially replicated'). The effectiveness of these survey methods at detecting species richness and rare species were tested in Everglades National Park, Florida, USA.

| Survey method | Advantages | Disadvantages |
|---|---|---|
| Mobile driving transect surveys | - Allows estimates of bat abundance<br>- Enables sampling of a large spatial area in one night<br>- Allows sampling of areas that may be inaccessible for other survey techniques (e.g., private lands)<br>- Low equipment cost (e.g., requires only one detector) | - Introduces spatial sampling biases (e.g., some bat species may avoid or be attracted to roads)<br>- Provides limited spatial sampling (sampling is restricted to locations & habitats where roads are constructed)<br>- Provides limited temporal sampling (e.g., transects are typically surveyed only a few hours per night)<br>- Introduces temporal sampling biases (areas along the transect are sampled only briefly)<br>- Labor intensive (e.g., requires an individual to drive throughout the duration of the survey) |
| Stationary point surveys | - Low investment of labor (equipment is left to sample many hours after deployment)<br>- High temporal sampling replication (equipment can record all night for multiple nights) | - Does not allow estimates of bat abundance<br>- Provides limited spatial sampling (only one location is sampled per detector) |
| Single strategic | - Enables purposeful maximization of bat detections (e.g., if detectors are placed near water or other areas expected to have high bat activity levels)<br>- Low investment of labor (equipment is left in one place permanently) | - Provides limited spatial sampling (does not capture landscape heterogeneity) |
| Spatially replicated | - Provides spatial replication (captures landscape heterogeneity) | - Labor intensive (requires deployment of equipment at multiple sites) |

community. Further, mobile transects typically only sample a few hours after sunset each night for a limited number of nights each season (*Loeb et al., 2015*), and, because the vehicle is moving, sample only briefly at any given location. These sources of limited temporal and spatial sampling inherent to mobile survey efforts could lead to underestimates of species richness (*Skalak, Sherwin & Brigham, 2012*), and an inability to detect trends in rare species activity or abundance (*Jones et al., 2013*).

Species richness is widely used as a metric for biodiversity surveys (*Purvis & Hector, 2000*), monitoring programs (*Yoccoz, Nichols & Boulinier, 2001*) and prioritizing conservation areas (*Howard et al., 2000*). Identification of the most effective survey methods and sample effort to detect species richness across landscapes is prudent given limited conservation resources. To assist biologists who must select a single survey method, we compared the relative effectiveness of two commonly used, but rarely compared, acoustic survey methods (mobile transect and stationary surveys) at detecting bat species richness in a vast protected area suspected to contain rare and endangered bat species (Everglades National Park (ENP)). We then focused on two types of stationary surveys to assess how spatial replication (surveys at multiple sites) influenced estimates of species richness within two structurally different vegetation communities. Specifically, we compared: "single strategic stationary surveys"—a single detector strategically located near an open water source to maximize bat detections, to "spatially replicated stationary surveys"—multiple detectors distributed across each vegetation community to capture spatial heterogeneity

in the bat community. Finally, we estimated the minimum required temporal sampling effort for each method to detect maximum species richness. Seasonal flooding, limited road access and poor knowledge of the bats in ENP make it challenging to design an effective bat monitoring protocol, yet is representative of many remote, conservation areas worldwide.

## MATERIALS AND METHODS

All field methods for this study were approved by the United States Department of the Interior, National Park Service, Everglades National Park (permit number: EVER-2015-SCI-0009).

### Study area

This study was conducted in ENP, situated at the southern end of the vast freshwater wetland of the Greater Everglades Ecosystem in South Florida, U.S.A. ENP encompasses 610,484 ha of federally protected lands and is designated an International Biosphere Reserve, a World Heritage Site and a Wetland of International Importance in the Ramsar Convention (*Maltby & Dugan, 1994*). ENP is characterized by two distinct seasons: dry/cool (December to April) when average monthly precipitation (42–47 mm) and average temperatures (12–25 °C) are relatively low, and wet/warm (May to November) when the majority of the annual precipitation falls (62–200 mm per month) and average temperatures are high (18–32 °C) (*Duever et al., 1994*; *Lodge, 2010*). The three dominant vegetation communities in ENP include pine rocklands (hereafter 'pinelands'), freshwater sawgrass marshes and wet prairies (hereafter 'prairies'), and mangrove swamps (hereafter 'mangroves') (*Florida Natural Areas Inventory, 2015*; *Lodge, 2010*). Access is greatly limited by the restriction of terrestrial motor vehicle access in the park to one paved road, and also the seasonal flooding that extends across much of the park.

Very little is known about bat community composition in ENP (*Lodge, 2010*). Based on range maps (*Florida Fish and Wildlife Conservation Commission, 2016*; *IUCN, 2015*), at least seven species of bats likely occur in ENP: big brown bats (*Eptesicus fuscus*), Florida bonneted bats (*Eumops floridanus*), Seminole bats (*Lasiurus seminolus*), northern yellow bats (*L. intermedius*), evening bats (*Nycticeius humeralis*), tri-colored bats (*Perimyotis subflavus*), and Brazilian free-tailed bats (*Tadarida brasiliensis*). Four of these species (tri-colored bats, Florida bonneted bats, Seminole bats, and big brown bats) are considered to be uncommon or rare in ENP (*Florida Fish and Wildlife Conservation Commission, 2016*), and one is of particular concern (Florida bonneted bat), due to its status as endemic to south Florida and Federally Endangered (*United States Fish and Wildlife Service, 2013*). Four additional species have geographic ranges within south Florida that could possibly extend into the park: velvety free-tailed bats (*Molossus molossus*), southeastern myotis (*Myotis austroriparius*), eastern red bats (*L. borealis*) and Rafinesque's big-eared bats (*Corynorhinus rafinesquii*).

### Acoustic surveys

We implemented three types of acoustic survey methods (one mobile and two stationary) that took into account the major vegetation communities and accessibility to survey

locations: (1) mobile transects, (2) single strategic stationary surveys, and (3) spatially replicated stationary surveys (Fig. 1A). We selected ultrasonic recording equipment designed and recommended for use in stationary and mobile acoustic monitoring studies (*Loeb et al., 2015*). Stationary surveys were conducted with full spectrum Song Meter SM3BAT detectors and external ultrasonic microphones (SM3-U1; Wildlife Acoustics, Inc., Maynard, MA, USA) extended to 3 m above ground level. We conducted mobile transects using a full spectrum Echo Meter EM3+ detector and external ultrasonic microphone (SMX-UT; Wildlife Acoustics, Inc.) mounted on the top of our vehicle, and connected a Global Positioning Satellite (GPS) unit to the detector to record GPS coordinates for each acoustic file. Prior to deployment of all equipment, we used an Ultrasonic Calibrator (Wildlife Acoustics, Inc., Maynard, MA, USA), designed to verify the sensitivity of the microphones and overall system performance. We collected data with the three methods within a 40-night sampling period (details below) in each of two distinct seasons during 2015: dry/cool (February–March) and wet/warm (June–July).

### Method 1: mobile transect surveys

We used the only existing paved park road from the main ENP entrance gate to Flamingo (64 km), which traverses the three vegetation communities: pinelands, prairies and mangroves (Fig. 1A). Transects were driven at a constant 32 km/h, beginning 30 min after sunset (*Loeb et al., 2015*; *United States Fish and Wildlife Service, 2012*), for a survey duration of two hours. We surveyed the transect nine times in the dry season and ten times in the wet season, each within a 40-night sampling period (ca. once every five nights), alternating the start location between each end of the transect road for each survey. We selected a sample effort much greater than the two survey nights per maternity season recommended by the widely-implemented NABat protocol (*Loeb et al., 2015*), to estimate the number of survey nights necessary to detect maximum species richness in the study area. We did not survey on nights when average wind speeds exceeded 24 km/h (*United States Fish and Wildlife Service, 2012*), but allowed for wind gusts up to 30 km/h due to consistently high coastal wind conditions. Due to frequent but often short-duration rain events in south Florida, we paused each survey when precipitation began and waited for it to pass. If the survey was paused for more than 15 mins cumulatively, we ended the survey and repeated it the following night. We assigned each bat detected during a mobile transect to the vegetation community in which it was recorded, using that file's associated GPS coordinates (ArcMap v10.2.2; Florida Cooperative Land Cover Map v3.1).

### Method 2: single strategic stationary surveys

We established one easily accessible, strategically-located stationary monitoring station designed to remain permanently in each of the three vegetation communities: pinelands, prairies, and mangroves (Fig. 1A). To maximize bat detections and the quality of recordings, we located each detector within 10 m of a water source, but away from vegetative clutter and flat surfaces prone to acoustic echoes (*Britzke, Gillam & Murray, 2013*; *Loeb et al., 2015*). Detectors were powered by solar panels, mounted with custom made structures constructed from PVC pipes and angle iron, and programmed to record from 30 mins

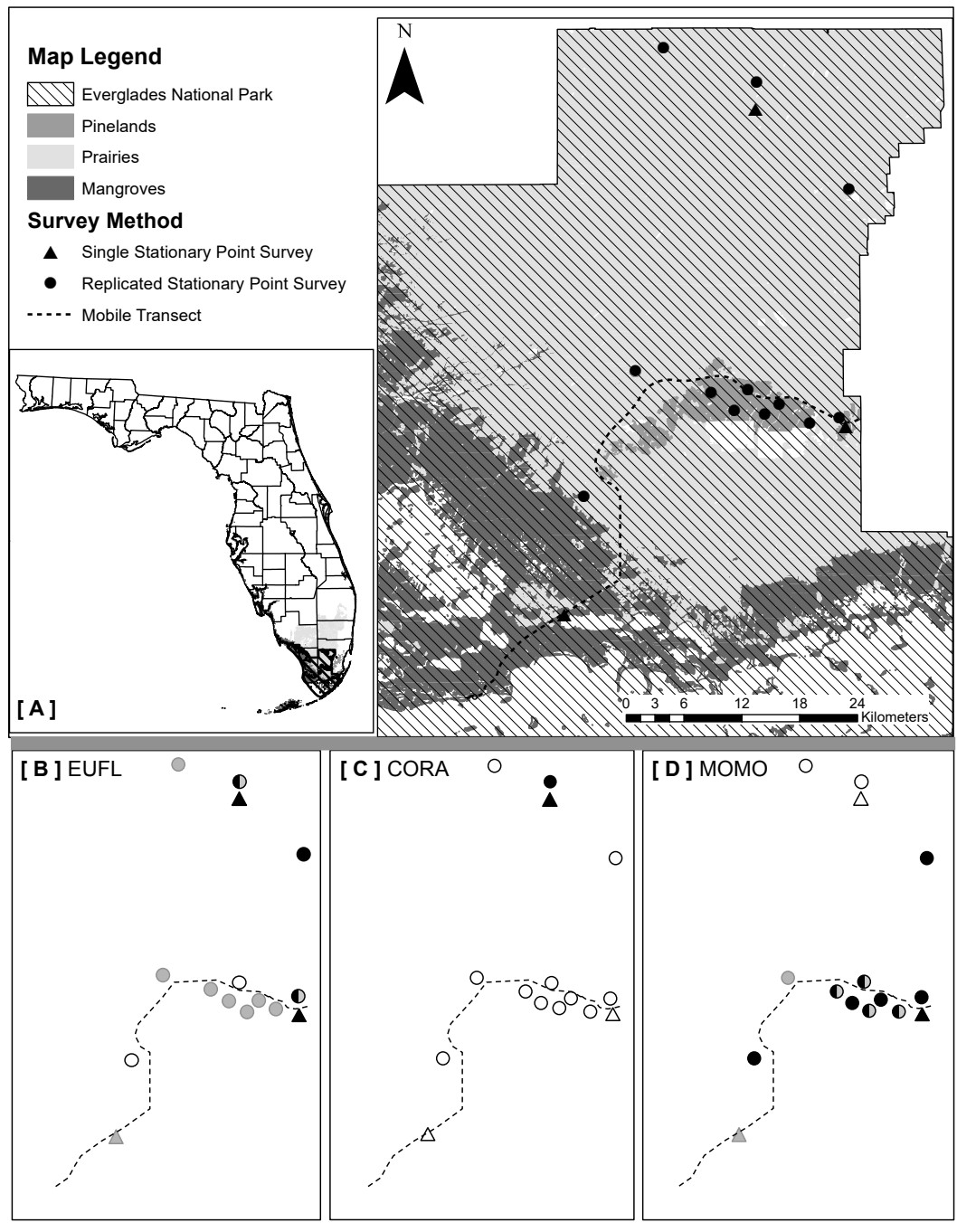

**Figure 1** (A) Map of acoustic detector locations for three survey methods across Everglades National Park, Florida, USA; created with GIS data from the Florida Natural Areas Inventory (FNAI 2015; Cooperative Land Cover Map, v3.1). (B–D) show locations of stationary detectors where rare bat species (EUFL, *Eumops floridanus* (B); CORA, *Corynorhinus rafinesquii* (C); MOMO, *Molossus molossus* (D)) were detected on the same nights and times when mobile transects were conducted (black symbols), detected on the same nights but at different times (e.g., later in the night) than when mobile transects were conducted (half-black symbols), detected on different nights than when transects were conducted (gray symbols), or not ever detected (open symbols). Mobile transects did not detect these three species during the study.

before sunset to 30 mins after sunrise (12-hr mean survey duration/night over the study). We set the detectors to record for 40 nights in each season.

### Method 3: spatially replicated stationary surveys

To assess how increasing sample effort spatially would affect estimated bat species richness and how that may differ between structurally distinct vegetation communities, we focused on pinelands (small, dry, closed canopy system) and prairies (vast, wet, open canopy system), two relatively easy to access vegetation communities. We did not sample in mangroves due to high water levels year round that would require additional equipment (e.g., airboats) for these surveys to be conducted. We sampled a total of six locations in pinelands (2–12 km apart) and six locations in prairies (8–45 km apart), distributing sample locations as evenly as possible throughout each vegetation community ($\geq$ 300 m from the habitat edge), while ensuring that each location was accessible by foot year round (10–300 m from a secondary road or path, 100–3,500 m from a primary road) (Fig. 1A). We mounted detectors and microphones on painter's poles (3 m above ground) and selected sites where the recording space above microphones was unobstructed by vegetative clutter. We programmed detectors to record from 30 mins before sunset to 30 mins after sunrise (12-hr mean survey duration/night over the study). In each season, we conducted two 20-night sampling sessions. During each 20-night sampling session we placed three detectors in pinelands and three detectors in prairies and then moved these detectors to six new locations for the second 20 night sampling session. The same 12 locations were surveyed during both the wet and dry seasons.

## Acoustic analysis and species identification

We used Kaleidoscope Pro 3.14B (Wildlife Acoustics, Inc.) for automated noise (non-bat ultrasound) filtering, species classifications and to manually review the spectrograms of acoustic files. We used the 'Bats of Florida Classifier' (beta v.3.1.0) and 'Bats of the Neotropics' (v.3.1.3) in Kaleidoscope Pro to identify calls from all possible species in the region. Species with indistinguishable calls (eastern red bats and Seminole bats) were considered as one species group. Current classifiers in Kaleidoscope Pro can not identify Rafinesque's big-eared bats; hence, we used Townsend's big-eared bats (*Corynorhinus townsendii*) as a proxy, given the similarity in call structure between the two species (*Szewczak, 2011*). To minimize species identification error and false positives (*Lemen et al., 2015*), we developed a conservative protocol to positively identify species with infrequent detections and to filter out ambiguous and/or low quality bat calls. We required that all files that were given an ultimate species classification had $\geq$ 5 total calls and that $\geq$ 75% of the calls in each of these files matched the ID assigned by Kaleidoscope Pro. To confirm species identifications of files that met our criteria, we manually reviewed: (1) any unexpected species classifications based on species range maps, (2) any species with <10 files at a given detector location and (3) all files identified as southeastern myotis, velvety free-tailed bats or Rafinesque's big-eared bats due to distribution uncertainty for these species. Finally, due to the status of Florida bonneted bats as an endangered species and their easily identifiable echolocation calls, we manually reviewed 100% of the files classified by Kaleidoscope Pro

as Florida bonneted bats, NoID, or Noise. We also reviewed all ambiguous files for which the software provided multiple species identifications including Florida bonneted bats (i.e., Florida bonneted bat was listed as an 'Alternate' species). We only included files that we unambiguously identified as Florida bonneted bat calls and did not overlap with Brazilian free-tailed bats' frequency range (<18 khz). To aid in manual validation, we compiled a reference library of echolocation calls from hand-released bats captured in mist nets in south Florida and other areas in the southeastern US, as well as from publicly available databases and journal articles (e.g., *Jung, Molinari & Kalko, 2014*; *Marks & Marks, 2006*; *Szewczak, 2011*). To reduce subjectivity, two researchers experienced in identification of bat calls independently confirmed all manually validated calls. Our automated and manual validation protocols were used consistently across detectors; thus, despite some expected identification error, we were able to compare the relative ability of each survey method to detect species richness.

## Statistical analysis

All graphical and statistical analyses were conducted in the statistical software R (v. 3.1.2) with R studio (v. 0.98.1102). All reported errors are standard error of the mean ($\pm$SE) unless otherwise noted. To evaluate the effectiveness of the three survey methods at representing bat community composition, at each detector we quantified: (1) total species richness (number of species) detected over the entire study, and (2) nightly species richness (number of species standardized by survey night to account for uneven temporal sampling among survey methods).

### *Survey method*

First, we did a simple comparison of the species (and total species richness) detected by all three methods in each survey location in each season. To account for the limited temporal sampling by mobile transects relative to stationary surveys, and to identify potential reasons for differences in species detections, we subsequently restricted this comparison to: (a) the nights, (b) the nights *and* times when mobile transects were conducted.

Next, we compared nightly species richness between mobile transects and single strategic stationary surveys only, to test the effectiveness of these two methods across the three primary vegetation communities. To account for the greater number of nights sampled by the stationary surveys, we limited our comparison to only the nights when mobile transect surveys were conducted. Although the number of recording hours per night was less for mobile transects than stationary surveys, we used 'survey night' as our sample unit based on existing survey protocols and what is typically used in the field (*Loeb et al., 2015*; *United States Fish and Wildlife Service, 2012*). In the practical application of these survey methods, mobile transects are never conducted all night, and stationary surveys are generally always conducted for the full night. We constructed generalized linear mixed-effects models (GLMMs), with vegetation community and season as random effects (function *glmer,* R package *lme4 Bates, Maechler & Bolker, 2012*). We modeled all GLMMs with a Poisson distribution for count data, after assessing model fit using standard graphical diagnostics of residuals (*Zuur et al., 2009*) and testing for over-dispersion (function *dispersion_glmer*, R

package *blmeco* (*Korner-Nievergelt et al., 2015*). To test for the effect of survey method, we compared two nested models (one with survey method included and one without) using a likelihood ratio test (function *anova*, R Stats package).

### Sample effort

To evaluate how spatially increasing sample effort within vegetation communities affected the estimated species richness, we compared the two stationary survey methods in pinelands and prairies and used the entire sampling period (40 nights in each season). We compared the nightly species richness that was detected by single strategic and spatially replicated stationary surveys. We constructed GLMMs (Poisson distribution), with survey method and vegetation community as fixed effects and season and individual detector as random effects (categorical variables), and used likelihood ratio tests to evaluate the effect of survey method. We then used the same approach to test for differences between survey methods within pinelands and prairies to determine if there was a difference in the effectiveness of spatial replication in two structurally distinct vegetation communities.

### Species accumulation curves

To further compare survey methods and to estimate the minimum temporal and spatial sampling effort necessary to detect maximum expected species richness, we generated sample-based rarefaction species accumulation curves (SAC) (*Gotelli & Colwell, 2001*), using the function 'specaccum' in the R package *Vegan*. We used the default 'exact' method, which estimates the expected species richness using a sample-based rarefaction method (*Chiarucci et al., 2008*; *Kindt, Vandamme & Simons, 2006*) and is often known as the Mao Tau estimate (*Colwell et al., 2012*). Maximum expected species richness is estimated at the point that SACs reach an asymptote, or the probability of detecting new species approaches zero. Rarefaction curves can be used to compare the efficacy of different sampling methods within an area(*Ellison et al., 2007*; *Longino, Coddington & Colwell, 2002*). We used a simple, but approximate, method to compare SACs among survey methods by looking at the degree of overlap in confidence intervals (CI) (*Gotelli & Colwell, 2011*). To do this, we calculated unconditional standard deviations (*Colwell, Mao & Chang, 2004*; *Colwell et al., 2012*), which are not conditioned on the empirical data set but are instead based on an estimation of the extrapolated number of species in the regional species pool (i.e., assumes the sample is randomly drawn from a larger, unknown, species assemblage (gamma diversity); function *specpool*). We then plotted 84% confidence intervals ($ci = 1$ in function *specaccum*) for each SAC, which are more appropriate for tests of overlap than traditional 95% CI, assuming approximately equal CIs (*Gotelli & Colwell, 2011*; *Payton, Greenstone & Schenker, 2003*). To compare SAC overlap between single strategic stationary surveys and mobile transects, we limited the sample effort to the same number of nights for each method by generating a random subsample, without replacement, of single strategic stationary survey nights (nine in the dry season and 10 in the wet season). To determine the temporal sample effort required to detect all bat species using each survey method in each of the two seasons, we used the full sample period and plotted SACs with number of survey nights as the sample unit. Multiple detectors within each survey method (single strategic $= 3$; spatially replicated $= 12$) were pooled for these analyses. To assess how required

temporal sample effort varied by vegetation community, we plotted SACs within pinelands and prairies for the two stationary survey methods. Finally, to determine the minimum spatial sample effort required to detect all bat species using the spatially replicated stationary surveys in each season, we plotted SACs in pinelands and prairies with number of detector sites as the sample unit.

### Estimated survey effort and costs

To provide biologists with a relative comparison of the labor and expenses associated with each survey method in our study, we estimated the total *Effort* (person-hours to the nearest hour) and *Cost* (to the nearest $10) required for data collection and analysis for one season (40-night sampling period). We estimated *Effort* required for fieldwork (driving, deploying/removing equipment, changing batteries/data cards), and analysis (automated species ID and manual validation, but not statistical analyses). We estimated *Cost* required for equipment (detectors, microphones, batteries, data cards, acoustic analysis software), vehicle (current State of Florida vehicle reimbursement rate of $0.45/mile), and personnel (technician pay rate of $12.00/hr). To account for uneven temporal sample effort among survey methods, we then standardized the estimated costs by the number of survey nights.

## RESULTS

During one dry/cool and one wet/warm season, we recorded acoustic data from 720 detector nights (40 nights * 3 single strategic detector sites * 2 seasons; 20 nights * 12 spatially replicated detector sites * 2 seasons) with stationary surveys and 19 detector nights (19 nights * 1 detector) with mobile transect surveys. Due to equipment malfunction, data were not useable from one night of mobile transect data in the dry season and ten nights of data in both seasons for the single stationary detector in the prairie. We recorded 193,252 total acoustic files. After removing files determined to be non-bat ultrasonic or acoustic noise (e.g., wind, insects, birds), 151,259 files containing bat sequences remained for analyses (single strategic = 76,521; spatially replicated = 73,461; mobile transect = 1,277). Using our conservative protocol and manual validation, we identified 42,669 files (28.2% of total bat files) to the level of species or species group (single strategic = 16,384; spatially replicated = 26,054; mobile transects = 231). In total, we detected nine bat species (eight species and one species group): Rafinesque's big-eared bats, big brown bats, Florida bonneted bats, eastern red bats and/or Seminole bats, northern yellow bats, velvety free-tailed bats, evening bats, tri-colored bats, and Brazilian free-tailed bats. The species richness detected varied by survey method, vegetation community and season (Table 2).

### Survey method

Both single strategic and spatially replicated stationary surveys detected all nine bat species, whereas mobile transect surveys did not detect three of these species at any point during the study: Florida bonneted bats, Rafinesque's big-eared bats, and velvety free-tailed bats (Table 2). This finding was consistent even when we limited our comparison of survey methods to only the nights and the specific time window during which transects were conducted (Figs. 1B–1D). Further, mobile transects did not detect tri-colored bats or

**Table 2** Bat species suspected to occur in Everglades National Park, FL, USA that were detected (1) or not detected (0) by each survey method, vegetation community and season.

| Bat species | Survey method Overall | | | Vegetation Mangrove | | Pineland | | Prairie | | Season Dry | | Wet | |
|---|---|---|---|---|---|---|---|---|---|---|---|---|---|
| | SS | SR | MT | SS | MT | SS | MT | SS | MT | SS | MT | SS | MT |
| *Corynorhinus rafinesquii*[a] | 1 | 1 | 0 | 0 | 0 | 0 | 0 | 1 | 0 | 0 | 0 | 1 | 0 |
| *Eumops floridanus*[b] | 1 | 1 | 0 | 1 | 0 | 1 | 0 | 1 | 0 | 1 | 0 | 1 | 0 |
| *Eptesicus fuscus*[c] | 1 | 1 | 1 | 1 | 1 | 1 | 0 | 1 | 1 | 1 | 1 | 1 | 1 |
| *Lasiurus intermedius*[d] | 1 | 1 | 1 | 1 | 1 | 1 | 1 | 1 | 0 | 1 | 1 | 1 | 1 |
| *L. borealis/L. seminolus*[e] | 1 | 1 | 1 | 1 | 1 | 1 | 1 | 1 | 1 | 1 | 1 | 1 | 1 |
| *Molossus molossus*[f] | 1 | 1 | 0 | 1 | 0 | 1 | 0 | 0 | 0 | 1 | 0 | 1 | 0 |
| *Myotis austroriparius*[g] | 0 | 0 | 0 | 0 | 0 | 0 | 0 | 0 | 0 | 0 | 0 | 0 | 0 |
| *Nycticeius humeralis*[h] | 1 | 1 | 1 | 1 | 1 | 1 | 1 | 1 | 1 | 1 | 1 | 1 | 1 |
| *Perimyotis subflavus*[i] | 1 | 1 | 1 | 1 | 1 | 1 | 1 | 1 | 0 | 0 | 0 | 1 | 1 |
| *Tadarida brasiliensis*[j] | 1 | 1 | 1 | 1 | 1 | 1 | 1 | 1 | 1 | 1 | 1 | 1 | 1 |
| Species Richness | 9 | 9 | 6 | 8 | 6 | 8 | 5 | 8 | 4 | 7 | 5 | 9 | 6 |

Notes.
SS, Single Strategic; SR, Spatially Replicated; MT, Mobile Transect.
[a] Rafinesque's big-eared bats.
[b] Florida bonneted bats.
[c] Big brown bats.
[d] Northern yellow bats.
[e] Red/Seminole bats.
[f] Velvety free-tailed bats.
[g] Southeastern myotis
[h] Evening bats.
[i] Tri-colored bats.
[j] Brazilian free-tailed bats.

northern yellow bats in prairies, or big brown bats in pinelands, whereas these species were detected across all vegetation communities by stationary surveys (Table 2). Significantly higher mean nightly bat species richness (number of species present per survey night) was detected by single strategic stationary surveys ($3.59 \pm 0.23$) than with mobile transect surveys overall ($2.02 \pm 0.12$; $\chi^2(1) = 19.46, p < 0.001$), regardless of vegetation community or season. Further, in both dry and wet seasons, the asymptotes and CIs of our species accumulation curves (SACs) indicated that single strategic stationary surveys detected higher expected species richness and accumulated species faster than did mobile transects, even when restricted to the same number of survey nights as mobile transects (Fig. 2); however, we note that the CIs for each SAC differed, which violates one assumption in the use of overlapping CIs to determine inequality between the two methods (*Payton, Greenstone & Schenker, 2003*).

## Sample effort for stationary surveys
### Spatial—survey locations
The same total bat species richness ($N = 9$) was detected with both stationary survey methods (Table 2). Spatially replicated stationary surveys detected significantly lower mean nightly bat species richness ($2.74 \pm 0.06$) than did single strategic stationary surveys overall ($3.60 \pm 0.10$; $\chi^2(1) = 8.16, p = 0.004$). This result was also found within pinelands

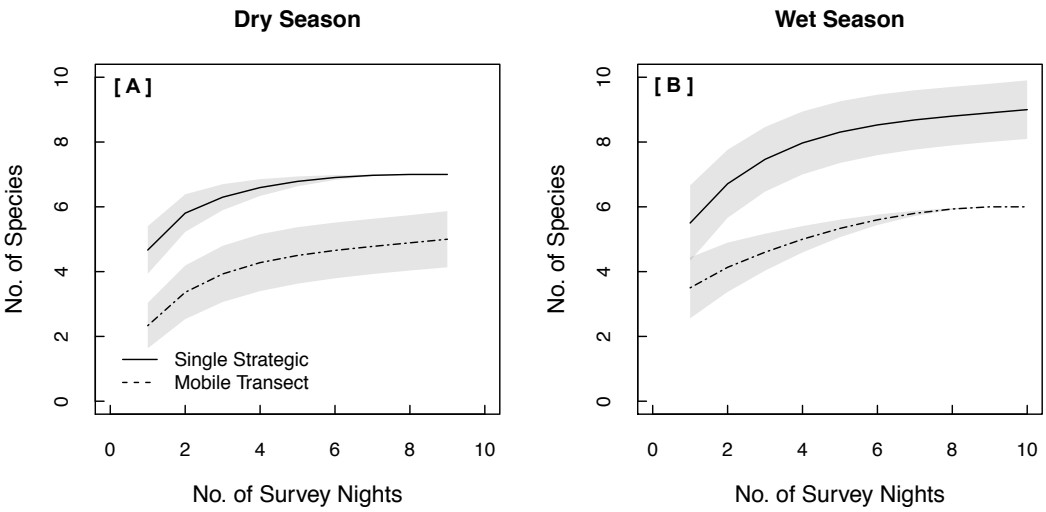

**Figure 2** **Species Accumulation Curves (SACs) across survey nights showing significant differences in expected species richness between single strategic stationary and mobile transect acoustic survey methods in both (A) the dry season and (B) the wet season.** To account for uneven sample effort between methods, this comparison was limited to the same number of survey nights used for mobile transects. Gray shading represents 84% confidence intervals (*Gotelli & Colwell, 2011*). All data were collected in 2015 in Everglades National Park, FL, USA.

$(3.23 \pm 0.09$ vs. $4.71 \pm 0.15$; $\chi^2(1) = 12.18$, $p < 0.001)$, but not within prairies $(2.24 \pm 0.06$ vs. $2.60 \pm 0.09$; $\chi^2(1) = 0.94$, $p = 0.33)$. In both seasons, the overlapping CIs of the SACs indicated that there was no significant difference in expected species richness or rate of species accumulation between single strategic stationary surveys and spatially replicated stationary surveys overall (Figs. 3A–3B) and within pinelands (Figs. 3C–3D). In contrast, within prairies in the dry season, the SACs of spatially replicated stationary surveys had higher expected species richness and steeper curves than the SACs from the single strategic stationary surveys (Figs. 3E–3F). In the wet season, overlapping CIs indicated no difference in expected species richness between single strategic stationary surveys and spatially replicated stationary surveys, but spatially replicated stationary surveys did not reach an asymptote within the 40-day sampling period indicating insufficient temporal sample effort to detect all species. When evaluating the optimal number of detector sites in spatially replicated stationary surveys, asymptotes were reached on the SACs with only four detector sites in pinelands during both the dry and wet season (Fig. 4). In contrast, spatially replicated stationary surveys in prairies did not reach asymptotes with the six detector sites in either season, indicating that spatial sampling effort was not sufficient.

*Temporal—survey nights*

The minimum number of sample nights required to detect expected species richness varied among survey methods, seasons and vegetation communities (Fig. 3). Maximum expected species richness ($N = 9$ species) was reached only during the wet season with stationary surveys in both pinelands and prairies combined (Fig. 3B). The minimum sample effort required to achieve this was 38 nights for both single strategic stationary surveys and for

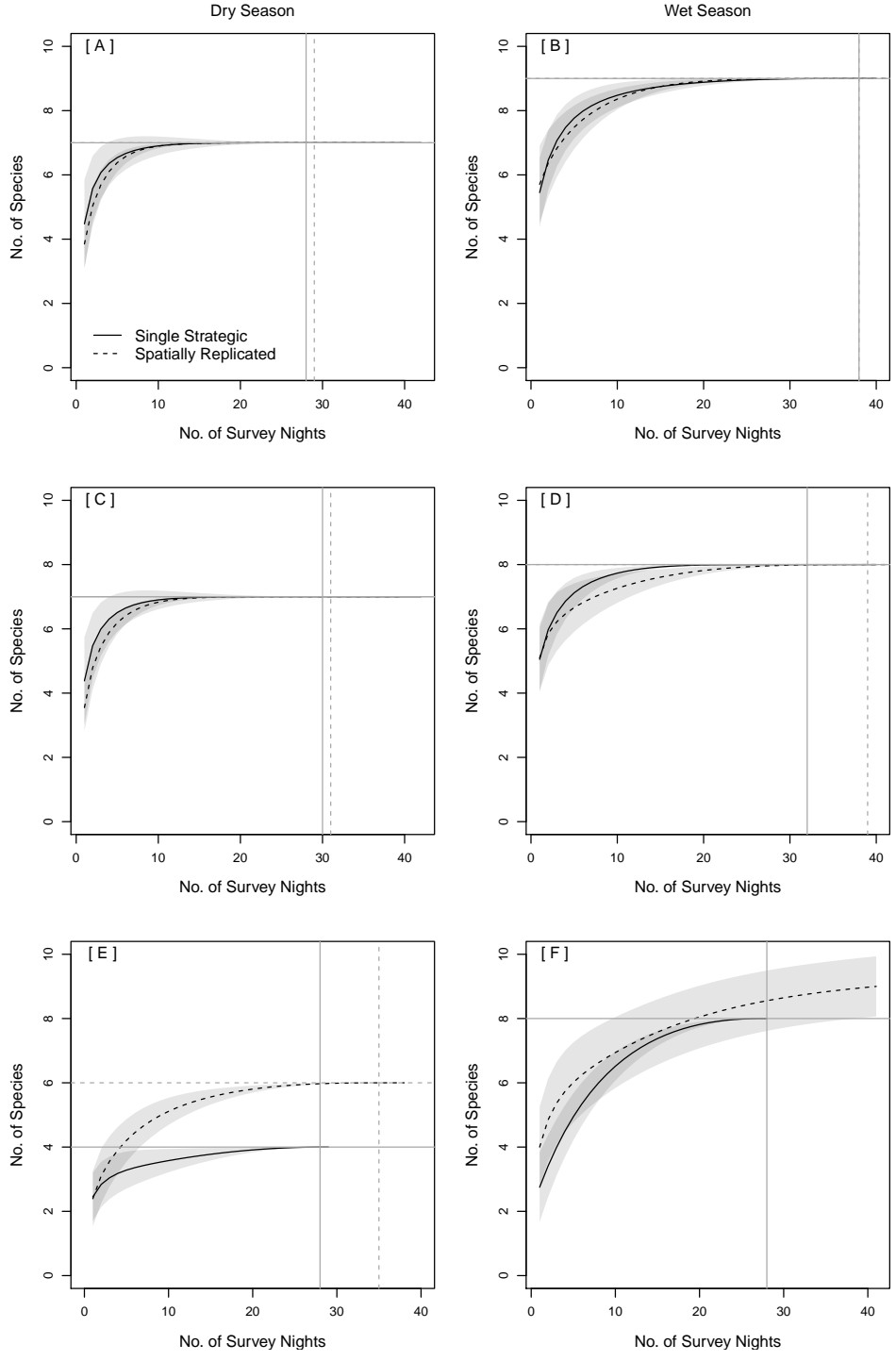

**Figure 3  Species Accumulation Curves (SACs) across survey nights showing differences between single strategic and spatially replicated acoustic survey methods: (A–B) overall (across both pinelands and prairies), (C–D) in pinelands and (E–F) in prairies.** Gray shading represents 84% confidence intervals (*Gotelli & Colwell, 2011*). Gray cross-bars indicate the point at which an asymptote was reached for each SAC, and the corresponding estimated species richness (*y*-axis) and sample effort (*x*-axis). All data were collected in 2015 in Everglades National Park, FL, USA.

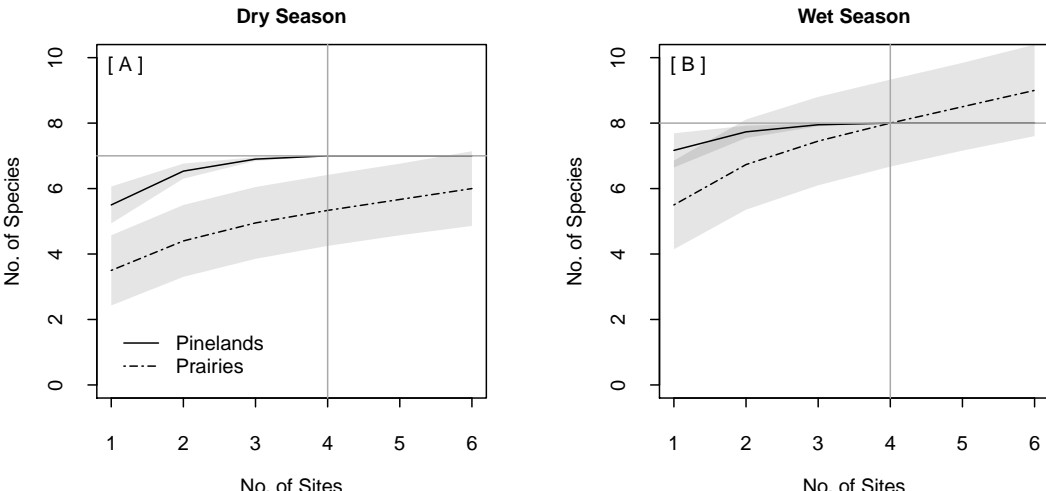

**Figure 4** **Species Accumulation Curves (SACs) for replicated stationary point acoustic surveys showing differences in the optimal number of detector sites between pinelands and prairies in (A) the dry season and (B) the wet season.** Gray shading represents 84% confidence intervals. Gray cross-bars indicate the point at which an asymptote was reached for each SAC, and the corresponding estimated species richness ($y$-axis) and sample effort ($x$-axis). All data were collected in 2015 in Everglades National Park, FL, USA.

spatially replicated stationary surveys. Although SAC asymptotes were also reached within the dry season and within each vegetation community, fewer than nine species were detected in each of these categories alone. An asymptote was not reached in prairies for spatially replicated stationary surveys in the wet season after 40 nights of sample effort (Fig. 3F).

## Estimated survey effort and costs

Based on our estimates of total effort and costs associated with the three survey methods, spatially replicated surveys required the most time and cost for collection and analysis of data, costing just over 1.5 times that of single strategic surveys and nearly four times that of mobile transects (Table 3). When standardized by the number of survey nights used in each method, the cost of mobile transects was the greatest per survey night and single strategic surveys the least.

## DISCUSSION

We showed that mobile transect surveys underrepresented bat species richness in ENP when compared to species richness detected with stationary surveys conducted on the same nights and during the same time of night. Most critically, mobile transects failed to detect three rare species over the course of the study: Florida bonneted bats (a federally endangered species), Rafinesque's big-eared bats (a species of greatest conservation need at the state level) and velvety free-tailed bats (a species not previously detected in the US outside of the Florida Keys). We also showed that increasing sample effort spatially across the landscape did not result in higher estimates of bat species richness overall, but did increase the rate at which new species were detected in one of two vegetation communities

**Table 3** **Comparison of estimated effort and costs among three acoustic survey methods used for one season (40-nights) of sampling in Everglades National Park, FL, USA.** Effort is reported as an estimated number of person-hours. Labor and analysis costs were derived from these estimates of effort.

| | | Mobile surveys | Stationary Surveys | |
| --- | --- | --- | --- | --- |
| | | **Mobile transect** *One 64 km transect driven every 5 nights; 10 transects total* | **Single strategic[a]** *Three sites sampled for 40 nights simultaneously* | **Spatially replicated** *Twelve sites sampled for 20 nights each; 6 sites at a time* |
| Effort (hours) | Field work | 45 h | 50 h | 80 h |
| | Analysis | 15 h | 30 h | 60 h |
| | Total effort | 60 h | 80 h | 140 h |
| Cost (usd) | Equipment[b] | $3,100 | $8,500 | $12,300 |
| | Vehicle[c] | $450 | $370 | $480 |
| | Field work[d] | $540 | $600 | $960 |
| | Analysis[d] | $180 | $360 | $720 |
| | Total cost | $4,270 | $9,830 | $15,460 |
| | Cost/Survey night[e] | $117 | $33 | $79 |
| Summary | Total effort | Low | Mid | High |
| | Total cost | Low | Mid | High |
| | Cost/Survey night | High | Low | Mid |

**Notes.**
[a] Estimates are based on all three stationary detector locations (mangroves, prairies, pinelands).
[b] Equipment cost includes detectors, microphones and acoustic analysis software.
[c] Vehicle cost was estimated based on rate of $0.45/mile.
[d] Field work and analysis time costs were based on a technician pay rate of $12.00/hour.
[e] Calculation excludes equipment costs, which do not change with number of survey nights.

(prairies but not pinelands). Finally, our results indicate that the temporal duration of stationary surveys needs to be long (>38 nights) to detect all bat species in the park, using our conservative identification protocol; thus, it may be more important to maximize temporal sample effort in a few locations rather than increase sample locations spatially.

## Survey method

Our results suggest drawbacks associated with the use of mobile transects as the sole survey method, particularly in regions with uncommon species and/or in large areas with limited road infrastructure. Given that protected areas worldwide tend to be in isolated locations, far from roads and with limited accessibility (*Joppa & Pfaff, 2009*), our findings raise concerns that mobile transects may not adequately represent bat communities in these areas. The failure of mobile transects in our study to detect the endangered Florida bonneted bat and two bat species previously undetected in ENP represents a particularly critical weakness of this method (*Arroyo-Cabrales & Álvarez Castañeda, 2008*; *Florida Fish and Wildlife Conservation Commission, 2016*; *Marks & Marks, 2006*; *United States Fish and Wildlife Service, 2013*). Furthermore, our finding that three other bat species (tri-colored, northern yellow and big brown bats) were not detected by mobile transect surveys within all three vegetation communities, as they were by stationary detectors, could result in erroneous conclusions regarding any changes in habitat use by these species.

Our results contradict *Whitby et al. (2014)*, who found that mobile transects detected the same species richness as stationary surveys and suggested that mobile transects may account for landscape scale variation across vegetation communities. Our conclusions may differ from *Whitby et al. (2014)* due to different activity patterns, habitat associations or species-specific abundances of the local bat assemblage sampled in each of our respective study regions. For example, abundant species, uniformly distributed across the landscape, would likely be detected by both survey methods, while rare or patchily distributed species may not. Similar to our study, *Whitby et al. (2014)* found that stationary surveys had steeper SACs than mobile transects, and thus required fewer survey nights to reach expected species richness. The need for greater temporal sample effort (survey nights or hours) to reach expected species richness represents an additional cost of mobile transects. The inability of mobile transects to detect rarer bat species in our study may have been due to limited temporal sampling (*Skalak, Sherwin & Brigham, 2012*), location of roads, possible avoidance of roads by bats (e.g., *Roche et al., 2011*; *Zurcher, Sparks & Bennett, 2010*) or a combination of these factors. Several stationary detectors only detected rare species on nights and at times (i.e., earlier or later in the night) when mobile transects were *not* being conducted, suggesting that mobile transects missed some rare bat detections due to their limited temporal sampling (see Figs. 1B–1D). However, these three species were also detected with several other stationary detectors on the *same* nights and during the *same* times when transect surveys were being conducted, indicating that these bats were present on the landscape while the survey vehicle drove the transect route, yet were still missed. Thus, the limited number of survey nights and hours was not the only reason these species were not detected during mobile transects. Rafinesque's big-eared bats were likely missed due to the location of the road on which the transect surveys were conducted; this species was only detected in the northern part of the park, 30 km from the transect road. In contrast, Florida bonneted bats and velvety free-tailed bats were detected by stationary surveys located near the mobile transect route (100–200 m of the road), yet never by mobile transect surveys. It is possible that these two species avoided the road on which we conducted the mobile transect; however, the road had very little traffic (<10 vehicles per survey) and no street lights, two features that have been found to negatively affect bat activity (*Stone, Jones & Harris, 2009*; *Zurcher, Sparks & Bennett, 2010*). We suggest that there was a spatio-temporal mismatch, in which species that are rare were missed due to the mobile detector only briefly recording in each given location. Similarly, *Stahlschmidt & Brühl (2012)* found that stationary detectors, when compared to walking transects, minimized the error associated with spatio-temporal variation in bat activity. Regardless of the reason, completely missing species with mobile transects is concerning given that a primary objective of acoustic surveys is to monitor trends in species and communities, and to inform conservation decisions.

Within a park the size of ENP (ca. 600,000 ha.), it is recommended that mobile transects be conducted in multiple locations (e.g., NABat suggests one for every 10 × 10 km grid cell); however, this would not be possible in our study area or much of the greater Everglades ecosystem due to limited roads and accessibility. We conducted mobile transect surveys on 19 nights, which, despite being a much greater sample effort than the two survey nights

recommended by NABat (*Loeb et al., 2015*), was not sufficient to detect all nine species detected with stationary surveys. We discourage the use of mobile transects as the primary method for long-term monitoring or assessing community composition of bats in large areas with minimal road access like we find in Everglades National Park. In the case of our study area, mobile transects did not provide any additional benefits to the data obtained from stationary surveys, thus did not warrant the additional cost and effort to conduct them (see Table 3). To circumvent some of the challenges associated with wildlife monitoring in large inaccessible conservation areas, use of alternative non-road based mobile surveys, such as Unmanned Aerial Vehicles (*Gonzalez et al., 2016*; *Linchant et al., 2015*) or balloons (*McCracken et al., 2008*), could be explored for bats. In marine systems, for example, new research suggests that autonomous underwater vehicles coupled with acoustic equipment may provide good coverage both spatially and temporally at a reduced cost compared to that of traditional survey techniques (*Klinck et al., 2016*). However, much more research would first be needed to determine how such methods might alter flight behavior of bats and subsequent interpretation of survey results.

## Sample effort for stationary surveys
### Spatial—survey locations

Contrary to what we expected, spatially replicated stationary surveys detected the same total species richness but lower mean species richness per night than single strategic stationary surveys, with no difference in the rate of species accumulation or estimated species richness overall. Various studies have shown that multiple detectors are necessary to adequately represent the diversity of bats across and within vegetation communities (*Ciechanowski et al., 2007*; *Duchamp et al., 2006*; *Froidevaux et al., 2014*; *Jung et al., 2012*; *Skalak, Sherwin & Brigham, 2012*). Our findings suggest that the more labor-intensive survey method of sampling spatially within vegetation communities was not necessary, and that a single detector, strategically located in pinelands and prairies to maximize bat detections, was sufficient. *Skalak, Sherwin & Brigham (2012)* suggested that detecting all bat species using a single detector was possible, but improbable in their study area (Mojave Desert, Nevada). Our placement of the single strategic detectors adjacent to bodies of water was likely responsible for the effective sampling of bat species richness. Although water is abundant in ENP, much of it is brackish and/or may be inaccessible to bats for drinking due to dense vegetation impeding flight paths. Because most bats require open sources of fresh water in order to swoop down to drink (*Kunz, 1982*; *Taylor & Tuttle, 2007*), bats in ENP may congregate around the limited open bodies of fresh water. Our findings support existing recommendations to select stationary survey sites that maximize potential bat activity, which are often near water (*Loeb et al., 2015*). Many ecosystems worldwide are either truly water limited for bats (e.g., xeric habitats, *Korine et al., 2016*) or may appear water rich yet be limited in sources of water that are accessible to bats (e.g., due to dense vegetative clutter, *Ciechanowski et al., 2007*; *Jackrel & Matlack, 2010*). In cases where there are no water bodies available for detector placement or there is limited knowledge of where bats are most active, spatial replication may be able to substitute for strategic placement of detectors.

Increased spatial sampling provided no additional benefits to estimates of species richness in pinelands but it increased the rate at which new species were detected and the maximum estimated species richness detected in prairies. Because prairies are open and wet (*Lodge, 2010*), a given open water source in prairies may be relatively less attractive to bats than an open water source in the drier, structurally complex pinelands. Thus, sampling near water in only one location in the prairies may not be sufficient to fully capture species richness in this community. Similarly, when evaluating the optimal number of detector locations and distribution needed to spatially represent bat communities in ENP, we again observed differences between the two vegetation communities sampled. Our finding that maximum species richness was reached (SAC asymptote) in pinelands with four sites (for 20 nights each) indicates that we adequately sampled spatial heterogeneity in this vegetation community in both dry and wet seasons. Although we found that the single strategic detector was sufficient to detect maximum species richness in pinelands, our estimates of minimum spatial sample effort for pinelands are applicable to monitoring schemes in other similar woodland communities that may not contain (or have access to) sources of water at which to locate detectors. In contrast, spatially replicated stationary surveys in prairies did not ever reach an asymptote in either season, indicating that more than six detectors (for 20 nights each) would be needed to fully represent the bat species richness in this vegetation community. *Skalak, Sherwin & Brigham (2012)* similarly found that at least six detector locations were necessary (for ca. 30 nights each) to detect 90% of bat species in a wetland system. We suggest that prairies in ENP required a greater sample effort than pinelands due to their much larger area and consequently higher heterogeneity in microhabitat and patterns of bat activity and species distributions, similar to findings by *Moreno & Halffter (2000)* of a greater sample effort required for bats in a large heterogeneous landscape than in small homogeneous habitats.

### Temporal–survey nights

We found that all nine bat species could be detected with two strategic stationary detectors (one in each of two vegetation communities) after 38 nights of sampling in the wet season when species richness was highest. In contrast, *Whitby et al. (2014)* detected all 12 bat species in their study area in Illinois after only 12 nights of sampling (three sample events of four nights each) with stationary detectors. A complete inventory of bat species can require a substantial sample effort (*Moreno & Halffter, 2000*). *Skalak, Sherwin & Brigham (2012)* recommended 20–30 sampling nights to detect 80–90% of the estimated bat species richness (12 species) in Nevada. *Froidevaux et al. (2014)* recommended a range of 12 to 33 nights for 90% of estimated species richness (16 species) in Switzerland. In our study, we detected 90% of the estimated species richness (eight species), after only seven nights of sampling with two strategically placed detectors in the wet season. The relatively low sample effort required to detect 90% of species in ENP, yet high sample effort required to detect all species is likely due to the presence of rare bat species, which have lower detection probabilities and higher variation in activity levels (*Skalak, Sherwin & Brigham, 2012*). The variation in optimal survey duration by these studies illustrates that blanket recommendations for monitoring protocols may not be

appropriate for all regions, and may need to be fine-tuned based on site-specific factors and project objectives.

## CONCLUSIONS

Our study highlights concerns with the use of mobile driving transect surveys as a short-term bat survey technique and long-term monitoring protocol. Rare species may not ever be detected with mobile transects or be detected too infrequently to provide meaningful data on population trends. Similarly, areas with limited roads—such as many national parks, wilderness areas, wildlife refuges, preserves and other vast conservation areas worldwide—may be particularly prone to road biases. Therefore, mobile transects may not accurately capture changes in bat populations/communities over time in these areas, which is the desired outcome of long-term monitoring. In contrast, stationary detectors at strategically selected locations may maximize bat detections and more accurately represent bat community composition and long-term patterns. This method is easily repeatable and requires a lower monetary investment per survey night than mobile transects. However, when conducting stationary surveys in areas containing rare species, such as in our study area, it is important to maximize temporal effort. We suggest that increasing the survey duration in a few strategic locations is a more time and cost-effective approach than increasing the number of survey locations, particularly in areas with accessibility limitations. Finally, our results demonstrate that the efficacy of survey methods can differ depending on the season and vegetation community. We emphasize the importance of field tests to optimize survey method, deployment duration/timing and spatial arrangement of detectors. By weighing survey project goals, resource availability and accessibility to sites, studies like ours will allow managers and researchers to make informed decisions before establishing long-term acoustic monitoring protocols for bats.

## ACKNOWLEDGEMENTS

We thank Tylan Dean and Skip Snow from Everglades National Park for coordinating on project design, development and logistics, and for providing valuable input throughout the project. We also thank Kirk Silas for assistance in the field and for valuable discussions about the manuscript.

### Funding

This work was supported by the United States Department of the Interior, National Park Service, Everglades National Park (Contract # P14AC01412). Publication of this article was funded in part by the University of Florida Open Access Publishing Fund. The funders had no role in study design, data collection and analysis, decision to publish, or preparation of the manuscript.

## Grant Disclosures

The following grant information was disclosed by the authors:
United States Department of the Interior, National Park Service, Everglades National Park: # P14AC01412.
University of Florida Open Access Publishing Fund.

## Competing Interests

The authors declare there are no competing interests.

## Author Contributions

- Elizabeth C. Braun de Torrez conceived and designed the experiments, performed the experiments, analyzed the data, contributed reagents/materials/analysis tools, wrote the paper, prepared figures and/or tables, reviewed drafts of the paper.
- Megan A. Wallrichs performed the experiments, prepared figures and/or tables, reviewed drafts of the paper.
- Holly K. Ober and Robert A. McCleery conceived and designed the experiments, contributed reagents/materials/analysis tools, reviewed drafts of the paper.

## Animal Ethics

The following information was supplied relating to ethical approvals (i.e., approving body and any reference numbers):

All research was passively collected and non-invasive via acoustics, thus no Institutional Review Board was necessary for this research.

## Field Study Permissions

The following information was supplied relating to field study approvals (i.e., approving body and any reference numbers):

All field methods for this study were approved by the United States Department of the Interior, National Park Service, Everglades National Park (permit number: EVER-2015-SCI-0009).

## Data Availability

The raw data and code have been uploaded as Supplemental Files.

## Supplemental Information

Supplemental information for this article can be found online at http://dx.doi.org/10.7717/peerj.3940#supplemental-information.

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
