# Peer review of "Mobile acoustic transects miss rare bat species: implications of survey method and spatio-temporal sampling for monitoring bats"

_PeerJ, doi:10.7717/peerj.3940_

## Round 0.1 · original submission · Minor Revisions

· Academic Editor

Minor Revisions

As a marine bioacoustician, I found your paper to be a helpful synopsis and comparison of methods being used to study bats. Both reviewers have requested only minor revisions, but in a wide range or areas (explanations, map clarity, statistics, etc.) I would only add to their comments that your discussion might be enlivened through consideration of the potential of non-road-based mobile transects (e.g. with drones?), and/or with any references you can make to similar comparative field studies with other species (e.g. marine mammalogists often tow hydrophone arrays behind ships, or more recently have used autonomous gliders to conduct mobile surveys).

Reviewer 1 ·

Basic reporting

It would be great to have a more thorough explanation for why you chose to do this methods comparison. You note that Whitby et al. 2014 found different results in their methods comparison. Can you explain in your hypotheses why you believed another method comparison was necessary in the ENP? E.g. was this because of the high species richness in the area? The diversity of vegetation, etc?

I really liked your Conclusion section. The rationale for the topic came through very clearly. Can you highlight some of those main questions in your introduction?

Experimental design

Page 4- Can you give a clear explanation for the benefits and issues of using each method type? It would help the reader understand the importance of your later findings to have this background. I would suggest creating a table showing: each method type, advantages/disadvantages, and the prominent literature citing recommendations. If there has not been adequate literature on one method this also highlights the importance for this and future methods studies. I think such a table would be a very useful/quick reference of what studies have previously suggested and been done.

The reason for testing the 2 methods against each other on Pg. 10 could use a better explanation. You have clearly set up what you aimed to quantify on Pg. 10 (line 203) in your methods. I would like to see this same clarity reflected in your results. Even by using the same subheadings later.

Some clarity on methods and site selection are missing. See my comments below.

Validity of the findings

I think you have found a useful result for researchers and conservation managers. It is nice to see the quantification of conservation methods.

Additional comments

Line 37- add to “document bat population trends…”

Line 69- Hayes 2000 – This ref is not in your citations. Please add.

Line 75 – Loeb 2015- This seems like it should be Loeb et al. 2015. Please change this throughout

Line 132- Be consistent with your use of “Figure 1” and “Fig. 1”

Line 137- I think I would add the make/model of the external microphone in parentheses here and then delete the next sentence starting with: “We conducted mobile transects using….” This just seems to be a bit wordy, but I recognize this is a stylistic recommendation.

Line 143- “We collected data with the three methods during a 40 night sampling period…”
Your methods here are a little bit vague. Can you add the number of hours of recordings per each method type?

Line 164- Can you explain more clearly how you selected your sites? For example, you used water sources, but was this randomly determined based on x distance to a water source? In easiy accessible areas? Near known roosts? Just a bit more detail here might better inform managers that want to follow this method in the future.

Line 177- “…accounting for accessibility…” does this mean locations were randomly selected within, for example, 10 m of a track? Might help to clarify this again.

Line 178 – Can you report the ~height of the painter’s poles?

Line 200- I would delete ‘language’ from this sentence since R is a programming language not a software language as such.

Line 215- Please clarify what ‘this analysis’ refers to.

Line 228- A lot of repetition of the word ‘test’ in line 228 and 229. Can you clean this a bit?

Page 11- Can you mirror the subheadings to your methods so that it matches the subheadings in your results? E.g. spatial…. Temporal

Page 13- You report the results of species richness based on season and vegetation in Table 1. I realize you are emphasizing the methods comparison, but it would be nice to see the difference in species richness based on these factors highlighted briefly at the end of PP1 on Page 13 and then reference Table 1 .

Line 333-35 – Interesting discovery and provides good info. for managers!

Line 348- please add “mobile” before ‘transect surveys’

Line 385 – “… monitors population trends…” I don’t think your research monitors population trends specifically at this point. Rephrase to rather emphasize the trends in species/community composition.

Line 449- To be more succinct, I would delete: “… some studies report the # of nights required to reach a specified threshold of the estimated species richness.”

Line 473- Can you give specifics on what “slightly more?”

Line 647- "pespective,,.." – Reference needs fixing

Line 657- "US Geological…"Fill in here.

Reviewer 2 ·

Basic reporting

- adequate
- a few mentions of "population trends", e.g. lines 65, 114 could be removed as this was not the intention of the study
- some difficulty locating sampling stations within Fig. 1 within darkest shading; perhaps larger map of ENP at expense of smaller image of Florida. More important to the manuscript to denote detail of study area than to know that ENP is in southern Florida.

Experimental design

- question of interest in the study is moderately well-described. There are a few shortcomings however. The given metric against which to measure estimates of species richness is derived from range maps. It is not obvious that range maps are completely accurate; furthermore, because two species cannot be uniquely identified acoustically (Eastern and Seminole), the "true" number of species is different than described by the range maps. Finally, the range maps, I presume, do not differentiate richness by habitat type. Consequently, the validity cannot be assessed at the level of habitat type, but rather, at the level of ENP. That creates a mismatch which makes it awkward for evaluating study design at the habitat level (level of spatial and temporal replication).
- it was not clear what randomisation scheme was used to place the spatial replicates. Given concern about distance between sampling stations and roads; I would have thought measures might have been taken to place replicate sampling stations perhaps away from roads; perhaps challenging in pineland habitat.
- unclear why there was no attempt to have spatial replication in mangrove habitat--manuscript should make some mention of this
- as noted in marginalia of the submitted PDF, the study could not readily be replicated because of the manner in which sampling locations were chosen and the inclusion criteria of Supplemental S1.

Validity of the findings

- data appear fine; I could replicate most of the analyses presented via the R script provided
- my main concern when species richness is the metric, maximal species richness is small (9) and species identification is auditory is the potential for false positive identification (recording presence of a species not detected). With so few species in the taxonomic target, only a single false positive might alter the inference of the study. Supplemental Appendix S1 discusses this, but the manuscript ought to highlight mitigation taken to prevent false positives and perhaps tabulate number of files falling into various eliminated (<5 calls; <75% calls with ID match, ...) categories.
- because of the absence of the mangrove habitat type from spatial replicates, this limits the scope of inference of this study, not just to ENP but to pineland and prairies of the park
- I made several comments regarding suitability of confidence interval overlap as an inferential tool as well as whether the R code using a multiplier of "2" on the SE produces the desired "84%CI"
- there exists a literature suggesting that CI overlap is not an effective means of examining differences. The "84%" level only resolves this matter when the SEs of the two contrasting estimates are equal, which is not the situation here. a closer reading of the supporting literature cited by Gotelli and Colwell (2011), namely Payton et al. (2003, J. of Insect Science 3:34) states "assuming equal standard errors (k = 1) yields γ = 0.166. In other words, if you wish to use confidence intervals to test equality of two parameters when the standard errors are approximately equal, you would want to use approximately 83% or 84% confidence intervals"
- Wolfe, R. and Hanley, J. (2002). If we’re so different, why do we keep overlapping? When 1 plus 1 doesn’t make 2. Canadian Medical Association Journal, 166, 65–66.
- Spanos, A. (2014). Recurring controversies about P values and confidence intervals revisited. Ecology 95: 645–651. doi:10.1890/13-1291.1
- this is not a large portion of the manuscript, more details included in the marginalia of the PDF, but it should be noted by the authors
- Lines 265-270 cause me concern. Is my logic correct? If so, deserves further discussion in the manuscript:
- line 266: note these two numbers of files are roughly equal (although there were double the number of spatially replicated nights)
- line 268: this step is unclear to me; <1/3 of files containing bat sequences were identified to species; is that the correct interpretation?
- line 269: whereas here the 16K to 26K ratio is roughly back into rough parity with the number of detector nights.
- What this suggests is that the "conservative protocol" eliminated a larger proportion of single strategic files than spatially replicated files.
-That seems as if it might be a threat to the comparison between the ability of two methods to detect bat species. If so, should that be addressed in the manuscript?

Additional comments

It is challenging to experiment with design parameters of a monitoring programme, nevertheless, an important component of any monitoring work is to determine whether it is able to achieve its goals. Hence, the goals of the programme need to be explicit and evaluation of the attainment of the goals is nedessary.

Annotated reviews are not available for download in order to protect the identity of reviewers who chose to remain anonymous.

---

## Round 0.2 · accepted · Accept

· Academic Editor

Accept

Great job responding to both reviewers' comments. I agree with Reviewer 1 that the new table is worth including. I am happy to accept the latest version of the manuscript as is, though you are welcome to respond to the final suggestions of Reviewer 1.

Reviewer 1 ·

Basic reporting

Suggested changes on basic reporting were addressed to my satisfaction or the authors provided logical reasoning for ignoring changes.

I like the addition of Table 1, because it seems like a great resource for methods. Is all information attributable to Loeb et al. 2015?

Check consistency around using ENP and 'Everglades National Park.' For example it is written out on line 455. Would suggest only writing out on first use.

Experimental design

Satisfied with changes addressed to make methods and questions more clear.

Validity of the findings

No Comment.

Additional comments

I am happy with the changes the authors have made to this manuscript. I think progress has been made to clarify some of the confusing aspects of project design, methods and the congruity with results. I look forward to seeing this paper contribute to the well-quantified studies that use sensory-based conservation/monitoring.